# Investigation of Surface Layer Condition of SiAlON Ceramic Inserts and Its Influence on Tool Durability When Turning Nickel-Based Superalloy

**Sergey N. Grigoriev** [ID]**, Marina A. Volosova** * **and Anna A. Okunkova** [ID]

Department of High-Efficiency Processing Technologies, Moscow State University of Technology "STANKIN", Vadkovskiy per. 3A, 127055 Moscow, Russia
* Correspondence: m.volosova@stankin.ru; Tel.: +7-916-308-49-00

**Abstract:** SiAlON is one of the problematic and least previously studied but prospective cutting ceramics suitable for most responsible machining tasks, such as cutting sophisticated shapes of aircraft gas turbine engine parts made of chrome–nickel alloys (Inconel 718 type) with increased mechanical and thermal loads (semi-finishing). Industrially produced SiAlON cutting inserts are replete with numerous defects (stress concentrators). When external loads are applied, the wear pattern is difficult to predict. The destruction of the cutting edge, such as the tearing out of entire conglomerates, can occur at any time. The complex approach of additional diamond grinding, lapping, and polishing combined with an advanced double-layer (CrAlSi)N/DLC coating was proposed here for the first time to minimize it. The criterion of failure was chosen to be 0.4 mm. The developed tri-nitride coating sub-layer plays a role of improving the main DLC coating adhesion. The microhardness of the DLC coating was $28 \pm 2$ GPa, and the average coefficient of friction during high-temperature heating (up to 800 °C) was ~0.4. The average durability of the insert after additional diamond grinding, lapping, polishing, and coating was 12.5 min. That is superior to industrial cutting inserts and those subjected to (CrAlSi)N/DLC coating by 1.8 and 1.25 times, respectively.

**Keywords:** ceramic inserts; diamond grinding defects; DLC-coating; nickel-based superalloy; polishing; SiAlON; surface layer; tool durability; turning





## 1. Introduction

Sintered tool ceramic based on $\alpha/\beta$ modifications of SiAlON is a more efficient solution for high-performance machining of high-temperature nickel superalloys such as Inconel 718 type alloys compared to hard alloys widely used for cutting inserts [1–4]. Nickel superalloy is one of the primary structural materials for manufacturing components of power equipment, aircraft engines, and spacecraft due to improved mechanical, anti-corrosion properties and structural stability at elevated operating temperatures. The improved mechanical properties of nickel alloys predetermine the increased heat and power loads on the tools that accompany machining and contribute to the intensification of the physicochemical interaction in the contact zones of the flank surface of the cutting inserts with the workpiece and the face surface with descending chips and high-intensity tool wear [5–9].

As production experience shows, nickel alloys begin to soften at temperatures corresponding to cutting speeds of 280–300 m/min and above, after which their mechanical processing is accompanied by significantly lower heat and power loads on the tool. In the specified high-speed range, the carbide tool instantly loses its cutting properties, while SiAlON ceramics are effectively used in turning operations due to higher heat resistance [10–13]. However, even the most modern brands of tool ceramics tend to cause brittle fracture of the cutting part during the nickel alloy turning at cutting speeds of more than 250 m/min with an increase in the cross-section of the cut layer (feeds of more than

0.15 mm/rev) with all the apparent advantages and, as a result, simultaneous exposure to significant thermal and mechanical loads on the tool [14–16]. High wear rates of the cutting part are observed under these conditions, which is associated with intensive friction and adhesion on the flank surface of the insert in contact with the workpiece and on the face surface in contact with the descending chips, even when choosing ceramic cutting inserts (CCI) with reinforced geometry and increased cutting-edge strength. At the same time, the worn section of the flank surface of the insert is characterized by the presence of craters with traces of adhesion, while relatively large recesses in the shape of holes are formed on the face surface, which often leads to micro-splitting and chipping of the cutting edge [17–21]. Therefore, with all the apparent advantages of CCI and the great potential of its use for critical product mechanical engineering, the actual share of its industrial use in the total global market volume of cutting tools is at a low level of about 11–12% [21,22].

Various researchers associate the low efficiency of CCI operation with the structural feature of tool ceramics, as well as the different volume and surface defects that form at the stages of the tool life cycle [23–25]. Among various defects, the critical role is played by defects of the surface layer since this layer primarily perceives operational loads, and the CCI efficiency depends on its condition [26–28]. The authors of this work propose the SiAlON CCI microstructural model presented in Figure 1. It explains the possible mechanisms of ceramic destruction under the influence of mechanical (P) and thermal (T) loads during cutting. The development of destruction of the surface layer is possible by one of three mechanisms:

1. intragranular destruction with gradual separation (abrasion) of microparticles of the surface layer;
2. grain-boundary destruction with separation of individual elements of the microstructure;
3. mixed destruction, in which there is a separation of grain conglomerates occurring inside the elements of the microstructure.

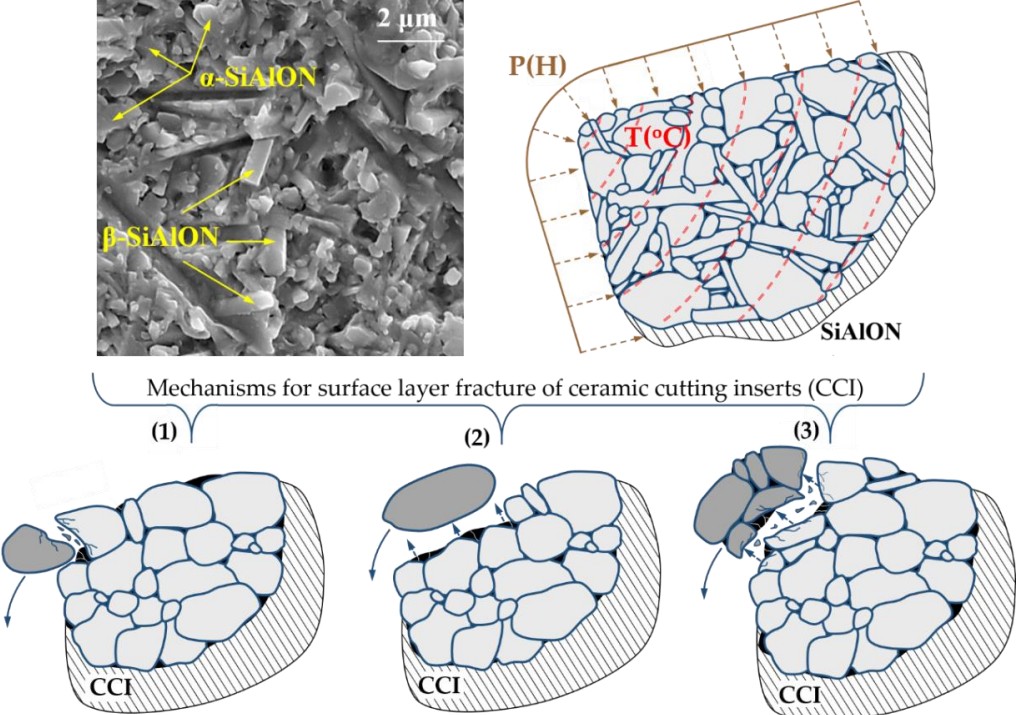

**Figure 1.** Variants of the destruction of the surface layer of SiAlON tool ceramics under external mechanical (P) and thermal (T) loads: intragranular (1), grain-boundary (2), and mixed (3).

The presence of numerous defects has a highly unfavorable effect on the resistance to the destruction of the surface layer under intense thermal and mechanical loads. De-

fects being stress concentrators contribute to the accelerated destruction of CCI contact surfaces and largely affect the stability of tool performance indicators. In particular, many authors have noted enormous variations in the durability (low operational stability) of cutting tools equipped with ceramic inserts, which hinders the industrial use of CCI in mechanical engineering.

The deposition of standard wear-resistant tool coatings cannot solve these problems since the increased defectiveness of the substrate significantly reduces the quality of the formed coatings and limits their functionality [29–33].

The authors of this work comprehensively investigated the influence of the condition of the surface layer of ceramic inserts made of $Al_2O_3$ + TiC, including the role of the formed wear-resistant coatings, on the performance of the tool when turning hardened bearing steels [28]. It was experimentally proven that the defects formed during diamond sharpening (grinding) of the ceramic inserts reduce the efficiency of the tool operation and contribute to premature destruction of the cutting part, which can occur at various stages of tool operation, in particular during the run-in period and at the stage of regular wear. It is possible to significantly minimize surface layer defects, the presence of which significantly affects the tool performance, by acting on the surface layer using various diamond abrasive machining methods [28,34–37].

The issues of the influence of the use of various diamond abrasive machining methods on the condition of the surface layer of SiAlON cutting tools and on the functioning of subsequently formed wear-resistant coatings on the performance of the tool when turning heat-resistant nickel alloys have so far remained out of the focus of attention of research groups. At the same time, these issues are highly significant from the point of view of a deeper understanding of the wear characteristics of SiAlON CCI and the scientifically justified use of various technological approaches to increase their performance, as well as more comprehensive industrial applications for machining heat-resistant nickel superalloys of the Inconel 718 type.

The purpose of the presented work was to study the influence of the surface layer condition of SiAlON ceramic cutting inserts processed by diamond grinding and polishing and subsequent (CrAlSi)N/DLC coating deposition on the tool operability in the high-speed turning of a heat-resistant nickel alloy with increased cross-sections of the cut layer.

The object of the research was the least previously studied SiAlON cutting insert. The focus of researchers until now has been mainly on square and rhombic plates made of $Al_2O_3$ + TiC ceramics [9,13,26,28], and less often $Si_3N_4$ [30,31]. At the same time, ceramic round plates made of SiAlON are designed for processing parts of gas turbine engines from heat-resistant chromium-nickel alloys. Along with titanium alloys, they are the primary structural materials for manufacturing responsible parts of aircraft gas turbine engines.

The novelty of this work lies in evaluating the surface layer condition influence on the operational ability of the ceramic cutting insert made of SiAlON cutting ceramic with the deposed double-layer coating of (CrAlSi)N/DLC structure, where the tri-nitride (CrAlSi)N sub-layer of the coating is responsible for better DLC coating adhesion knowing the problematic behavior of SiAlON to coatings, in conditions of extreme mechanical and thermal loads.

The practical significance of the work lies in evaluating the effect of diamond grinding and polishing in combination with the advanced double-layer (CrAlSi)N/DLC coating effect on the performance of industrially produced SiAlON ceramic cutting inserts in chrome–nickel alloy (Inconel 718 type) turning with increased cross-sections of the cut layer.

## 2. Materials and Methods

### 2.1. Cutting Tools, Material to Be Processed, and Laboratory Testing Methods

CRSNR 3232P 19 turning holders (Sandvik AB, Sandviken, Sweden) for external machining on universal lathes and CNC machines were used as a cutting tool for the research, in which RNGN-190800 round ceramic inserts of the AS500 brand manufactured by TaeguTec (Daegu, Republic of Korea) with a diameter of 19.05 mm and a thickness of

7.9 mm were mechanically fastened (Figure 2). When the insert is placed in the holder, the clearance angle is 5°, the rake angle is −5°, and the lead angle is 46°. It should be noted that the geometry of the cutting inserts was chosen from the tasks of providing curved surfaces for manufacturing gas turbine engine parts. SiAlON can be considered one of the most inconvenient materials for coating and machining that exhibits superior wear resistance compared to carbide tools, especially under extreme load conditions (semi-finishing in the temperature range up to 800 °C with increased depth of cut and feed). The tool material was SiAlON ceramic of the following composition: 79 vol% $Si_5AlON_7$, 17 vol% $Si_3N_4$, 4 vol% $Yb_2O_3$. The content of the main phases was revealed by X-ray diffraction analysis and processing of the results using the PANalytical HighScore Plus software by PANalytical B.V. (version 3.0) and the ICCD PDF-2 database (version 2023).

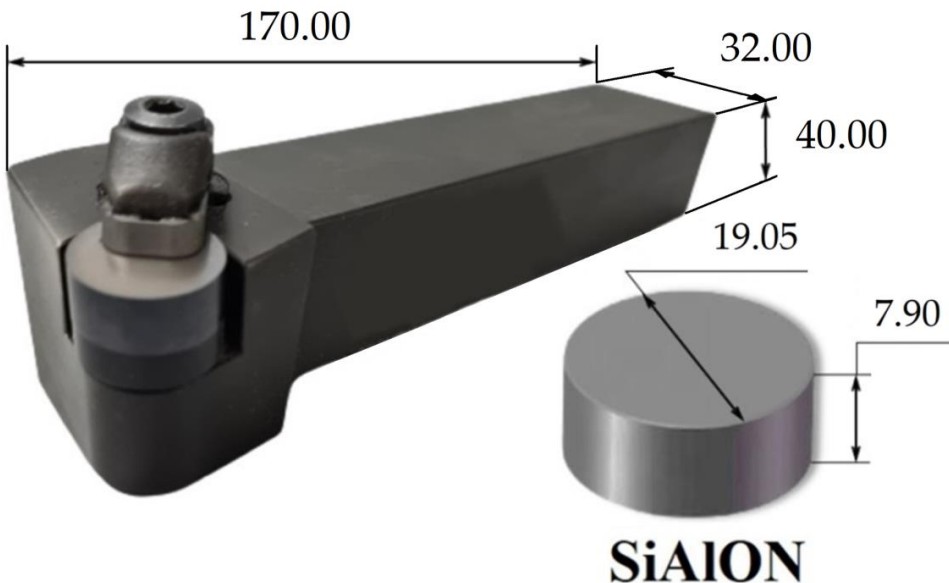

**Figure 2.** Design of a prefabricated turning cutter with mechanically fastened round ceramic inserts used in experiments.

XH45MBTJuBP nickel-based heat-resistant alloy, according to the national standard of the Russian Federation 5632-2014 (the closest analog of Inconel 718), was used as the material to be processed. Table 1 shows the composition of the XH45MBTJuBP alloy. The basis of this alloy is an austenitic solid solution of the nickel–chromium–iron system. This alloy is used for highly loaded elements of load-bearing structures and other parts of gas turbine engines operating in various climatic conditions at temperatures up to 800 °C. A hot-rolled bar with a diameter of 100 mm was used as a workpiece for the study. The nickel alloy had a hardness of 322 (HB) and a strength of 1130 MPa.

**Table 1.** Chemical composition of XH45MBTJuBP alloy used in the research.

| Element | Ni | Fe | Cr | Mo | Nb | W | Ti | Al | C, Si, Mn, S, P |
|---|---|---|---|---|---|---|---|---|---|
| Content (%) | 45.3 | 27.9 | 15.0 | 4.0 | 1.3 | 2.5 | 2.0 | 1.0 | 1.0 |

Workpieces were processed on a ZMM CU500MRD lathe (ZMM, Nova Zagora, Bulgaria) under cutting conditions providing intense heat–power loads on CCI: cutting speed V = 300 m/min, feed S = 0.2 mm/rev, and cutting depth t = 0.8 mm. Increased values were selected from the practice of semi-finishing difficult-to-machine alloys based on nickel and titanium. Increased feed at a simultaneously high cutting speed provides more intense heat and power loads on the tool when the probability of premature failure of the ceramic insert increases sharply. It is under these conditions that the role of the surface layer increases. It should be noted, within this study, that one value of the cutting speed was chosen so as not to overload the article with data and to conduct scientific groundwork for further research.

Ten CCI faces were tested to identify the nature of wear development over the cutting time. In order to minimize the error of experiments, all experiments were duplicated 10 times. The size of the wear area along the flank surface was measured every 2 min of turning on a Stereo Discovery V12 Zeiss optical microscope (Carl Zeiss AG, Oberkochen, Germany). The wear chamfer value of 400 μm was taken as the failure criterion.

When choosing the criterion of the flank wear, it should be noted that finishing of 0.3 mm is an optimum value, but only for tools that are subjected to regrinding. The cutting ceramic insert is not related to this group of cutting tools. They are much more expensive than hard alloy tools and cannot be reused after achieving the criterion. In most cases, this type of cutting tool is used carefully with a relatively small depth of cut. In this study, we used a relatively large depth of cut and feed to achieve the forementioned purposes to maximize tool life that is not subject to regrinding. Thus, the chosen cutting modes are not suitable for finishing but for semi-finishing, after which finishing is foreseen. In this condition, the tool is expected to develop its resource to the maximum. Therefore, flank wear of 400 μm is chosen.

### 2.2. Preparation of Cutting Ceramic Inserts with Different Condition of the Surface Layer

Industrially produced round-shaped SiAlON inserts were subjected to additional diamond abrasive machining operations such as lapping and polishing to form the experimental groups of ceramic inserts with different surface layer conditions. Additional processing of ceramic inserts was carried out on the Lapmaster Wolters lapping and polishing machine (Mt Prospect, IL, USA) with unique lapping and polishing wheels using various diamond suspensions (with a grain size of 50/40, 40/28 during lapping, and 10/7, 5/3 during polishing) with a cutting speed of 3 m/s. Finishing took 28 min, and polishing 16 min. Grinding and finishing modes were chosen based on literature data [35,37] and the authors' experience [9,16]. There were two groups of ceramic inserts made of SiAlON: industrial samples subjected to diamond grinding at production (I), experimental samples prepared in laboratory conditions of MSUT "STANKIN", subjected to additional lapping and polishing (II). Lapping and polishing as additional operations of abrasive machining of ceramic inserts were chosen because these processes can significantly minimize the degree of imperfection of the surface layer formed during diamond grinding, as noted by various researchers [34–37].

### 2.3. Coating of Ceramic Inserts

The choice was made in favor of a diamond-like (DLC) coating, which is deposited on a pre-formed nitride (CrAlSi)N sublayer based on the experimental data previously obtained by the authors of this study, when coating cutting tools designed for processing heat-resistant nickel alloys [38–43]. DLC coatings demonstrate the lowest coefficient of friction and depth of worn track when contacting the ball under high-temperature heating conditions during tribological tests and reduce the intensity of friction and adhesive interaction of the contact pads of the cutting tool when cutting nickel alloys [44,45].

DLC coatings, along with the described advantages, have specific features that should be taken into account when choosing the method of their deposition. First, there is a limitation of the maximum thickness of no more than 2.0 μm. Exceeding it leads to an increase in the level of internal stresses and an increase in the probability of their delamination when exposed to external loads [46,47]. Therefore, DLC coatings should be deposited on a preformed sublayer to increase the thickness of the formed coating on the cutting tool. In addition, the use of DLC coatings for high-speed cutting conditions of heat-resistant alloys is limited by their relatively low thermal stability. Formation of a thermally stable sublayer, for example, (CrAlSi)N, as well as doping of the DLC coating with various elements, for example, Si, allows the noted disadvantage to be minimized and ensures high efficiency of the DLC coatings [48–50].

The total thickness of the (CrAlSi)N/DLC coating that was formed on SiAlON CCI in the present study was 3.9 μm, including a 1.9 μm (CrAlSi)N sublayer and a 2.0 μm DLC coating.

It should be noted that the role of the (CrAlSi)N sublayer in cutting is in providing more favorable conditions for the functioning of the external DLC, increasing its adhesive bond strength to the substrate [10,41].

The technological process of (CrAlSi)N/DLC coating deposition was implemented on a multifunctional STANKIN unit (MSUT Stankin, Moscow, Russia), equipped with a set of systems and devices that provides purification of the processed samples and coating deposition by vacuum-arc evaporation of cathodes as well as chemical vapor deposition [51–55]. A schematic diagram and a general view of the unit are shown in Figure 3. The inner (CrAlSi)N layer was deposited by vacuum-arc evaporation of cathode materials, and the outer DLC layer was formed by the PACVD method by decomposition of hydrocarbon-containing gases in a gas discharge plasma. The DLC coating technology was optimized for ceramic tools, and the approaches developed by Platit AG (Selzach, Switzerland) were used as the scientific basis.

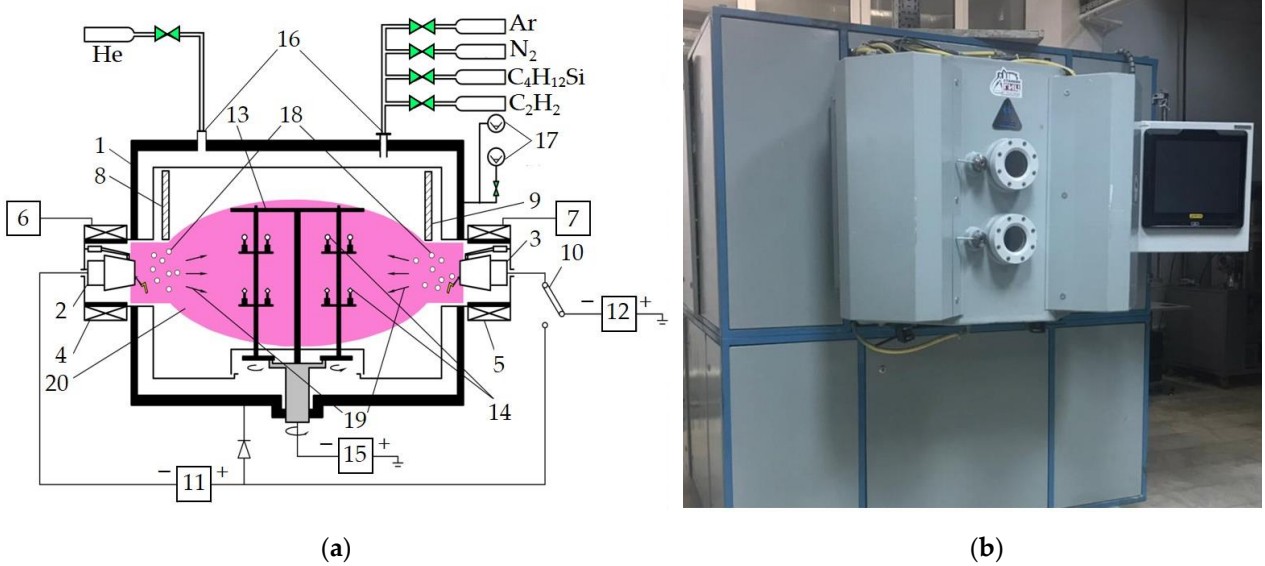

(**a**)　　　　　　　　　　　　　　　　　　　　　　　(**b**)

**Figure 3.** (**a**) Schematic diagram of the STANKIN technological unit for coating ceramic inserts, where 1 is vacuum chamber; 2,3 are cathodes (Cr and AlSi); 4,5 are cathode coils; 6,7 are power sources of cathode coils; 8,9 are cathode shutters; 10 is switchboard; 11,12 are cathode current sources; 13 is equipment for placing samples with planetary rotation; 14 is ceramic inserts; 15 is reference voltage power supply; 16 is gas supply systems; 17 is vacuum gauges; 18 is ions; 19 is ion movement directions; 20 is plasma; (**b**) a general view.

The complete technological cycle of coating deposition included six stages: sample heating, gas discharge purification, ion purification, a (CrAlSi)N sublayer deposition, a gradient DLC-Si layer deposition, and an external DLC layer deposition. Table 2 shows data on the modes that were assigned at each of the stages of coating formation. The modes mentioned were worked out in previous years and were chosen to ensure maximum adhesion of coatings [10,38,41].

### 2.4. Investigation of the Properties of the Surface Layer of Ceramic Inserts

A scanning electron microscopy method was used on Tescan VEGA3 LMH equipment (Brno, Czech Republic) for microstructural analysis of the surface of ceramic inserts with different surface layer conditions.

**Table 2.** Range of factors of technological cycle of coating deposition.

| Stage of the Process | Technological Modes | Measuring Units | Values |
|---|---|---|---|
| Sample heating | Chamber pressure | Pa | 0.03 |
| | Rotation speed of the tooling with samples (constant at all stages) | rpm | 5 |
| | Heating temperature | °C | 500 |
| | Heating time | min | 60 |
| Purification in a gas discharge | Composition of the working gas | - | Ar |
| | Chamber pressure | Pa | 1.2 |
| | Bias voltage | V | −650 |
| | Chamber temperature | °C | 500 |
| | Current at the cathode AlSi (open shutter) | A | 90 |
| | Current at the cathode Cr (closed shutter) | A | 105 |
| | Purification time | min | 20 |
| Ion purification | Composition of the working gas | - | Ar |
| | Chamber pressure | Pa | 2.2 |
| | Bias voltage | V | −800 |
| | Chamber temperature | °C | 500 |
| | Current at the cathode Cr | A | 90 |
| | Purification time | min | 20 |
| (CrAlSi)N sublayer deposition | Composition of the working gas | - | 95% $N_2$/5% Ar |
| | Chamber pressure | Pa | 0.9 |
| | Bias voltage | V | −40 |
| | Chamber temperature | °C | 500 |
| | Current at the cathode AlSi | A | 100 |
| | Current at the cathode Cr | A | 100 |
| | Deposition time | min | 90 |
| Gradient DLC-Si layer deposition | Composition of the working gas | - | 72% $N_2$/20% Ar/8% $C_4H_{12}Si$ |
| | Chamber pressure | Pa | 1.5 |
| | Bias voltage | V | −500 |
| | Chamber temperature | °C | 180 |
| | Deposition time | min | 20 |
| External DLC-layer deposition | Composition of the working gas | - | 55% Ar/45% $C_2H_2$ |
| | Chamber pressure | Pa | 0.8 |
| | Bias voltage | V | −500 |
| | Chamber temperature | °C | 180 |
| | Deposition time | min | 100 |

A Dektak XT stylus profilometer (Bruker AXS GmbH, Karlsruhe, Germany) was used to construct profilograms of the ceramic insert surface layer conditions. The specified device performs electromechanical measurements by contact scanning of the required surface area with a highly sensitive diamond tip at a given speed. Specialized software based on the analysis of measurement results processes information and visualizes the results in the form of 3D profilograms. Two parameters of the condition of the surface layer were evaluated, provided by the ISO 4287-2014 standard: $R_t$ is the sum of the most significant height of the profile peak and the most significant depth of the profile cavity (the total height of the profile by which the depth of the defective layer can be judged); $R_a$ is the arithmetic mean of the absolute values of the profile. According to the manufacturer's data, the measurement error is ±10% of the measured value when measuring from 100 μm up to 500 Å.

Crack resistance and microhardness of ceramic inserts with different surface layer conditions were determined on a universal QnessQ10A microhardness tester (Qness GmbH, Mammelzen, Germany) by the Vickers pyramid indentation method. The load on the indenter was 2 kg when assessing microhardness and 5 kg when assessing crack resistance. The indentation diagonals and the length of cracks propagating from the corners of the indentations were measured. Crack resistance ($K_c$) and microhardness (HV) were determined according to the known dependencies based on it [56].

An original method proposed by the authors of this study was used to assess the effect of surface defects on the ability of ceramic cutting edges to resist chipping when exposed to an external load. The approach described in [57,58] was taken as a basis. The possibility of obtaining new information about the mechanical behavior of ceramics when their edges are chipped by an indenter was substantiated by Griffiths' theory of brittle fracture. The authors of this study developed an improved method, the schematic diagram shown in Figure 4, taking into account the operational loads acting on ceramic inserts. The tests were performed on a Revetest scratch tester (Anton Paar, Corcelles-Cormondrèche, Switzerland) equipped with an acoustic emission (AE) signal registration sensor when exposed to a surface layer with a diamond indenter with a smoothly increasing load on the indenter (P) from 20 to 40 N. The diamond pyramid as an indenter was chosen purposefully to create the maximum stress concentration near the cutting edge. The identity of the location of the indenter application point relative to the cutting edge was strictly controlled. The chip had a shape close to a tetrahedron as a result of the force acting directly near the cutting edge. The experiments showed that the chip's length *B* and width *L* are not informative parameters for assessing the ability of the cutting edges to resist chipping (the chip size in all samples had similar values). The force at which the chip occurred turned out to be an indicator sensitive to changes in the condition of the surface layer of the ceramic samples. The moment of chipping and corresponding load were identified by the spectrum of the AE signal, which sharply increased at the moment of destruction. The high information content of the AE signal in the destruction of various materials has been noted in several works [59–63].

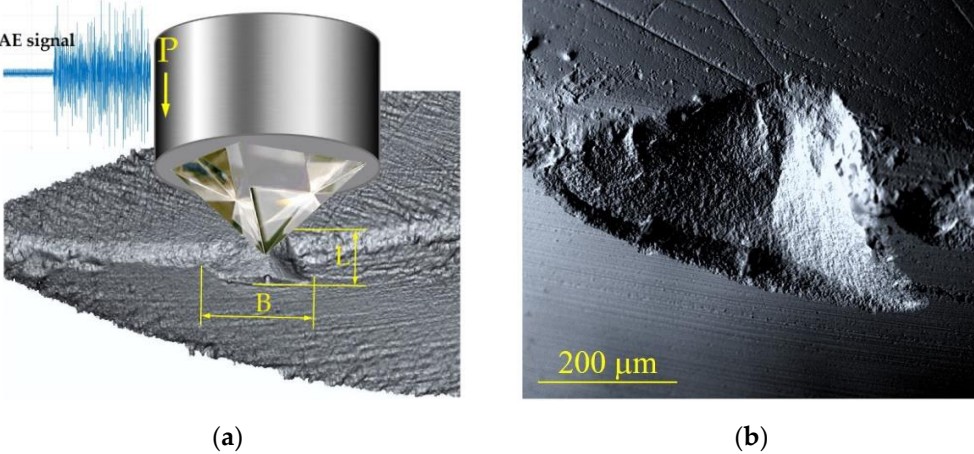

(**a**) (**b**)

**Figure 4.** Schematic diagram of the method used to evaluate the resistance of the CCI cutting edges to chipping: (**a**) the impact of the destructive load created by the diamond indenter near the cutting edge; (**b**) SEM-image of the chipping area of the cutting edge.

Tests were carried out on a Calowear CSM Instruments device (Peseux, Switzerland) under pressure on samples with a force of 0.2 N to study the ability of the surface layer of ceramic inserts to resist abrasion under abrasive conditions. The samples were affected by a rotating sphere of hardened steel when an abrasive suspension was supplied into the contact zone. Optical analysis of the wear holes' geometric dimensions and their measurement on a stylus profiler made it possible to quantify and qualitatively assess

the volume wear of samples [64]. The optical measurement error was calculated by the formula [65]:

$$\delta_l = \pm 3 + \frac{L}{30} + \frac{g \cdot L}{4000}, \tag{1}$$

$$\delta_t = \pm 3 + \frac{\Delta}{50} + \frac{g \cdot \Delta}{2500}, \tag{2}$$

where $\delta_l$ is the longitudinal measurement error, μm; $\delta_t$ is the transversal measurement error, μm; $\Delta$ is the measured length, mm; $g$ is the product height above microscope table glass (taken equal to zero), mm. The evaluation of the change in the coefficient of friction of ceramic inserts with different surface layer conditions was performed on a TNT-S-AH0000 tribometer (Anton Paar, Corcelles-Cormondrèche, Switzerland) when the ceramic insert rotates relative to a fixed ceramic ball with a diameter of 6 mm at a load of 1 N, a sliding speed of 10 cm/s and a test temperature of 800 °C [66].

The adhesion strength of the formed coatings with ceramic substrates by scratch testing with fixation of the spectrum of acoustic emission signals was evaluated on the NANOVEA M1 device (Irvine, CA, USA). The tests were performed with a linearly increasing load up to 50 N and a loading speed of 5 N/min. The acoustic emission spectra and corresponding forces were recorded during the test. The results of three measurements identified the normal load, which corresponded to the coating delamination moment, by [67–69].

## 3. Results and Discussion

### 3.1. Influence of Various Diamond Abrasive Machining Methods on the Condition and Characteristics of the Surface Layer of SiAlON Ceramic Inserts

The results of microstructural SEM-analysis of the surface layer of industrially manufactured CCI from SiAlON, subjected to diamond grinding at the finishing, show that the samples contain numerous surface defects (Figure 5), and the surface has a sophisticated morphological pattern. The surface layer of SiAlON tool ceramics has a specific relief, including a set of defects as a result of the impact of the diamond grains and the friction of the wheel binder on the surface of the high-density ceramics as well as local plastic deformation occurring during high-speed heating of the surface areas of the ceramics together with their rapid cooling. The observed defects (Figure 5) can be classified into the following types: (1) micro ridges, which are the result of plastic deformation, (2) ripped-out single grains and (3) ripped-out conglomerate grains, under force loads, (4) deep grooves, shaped by diamond grains of the grinding wheel, (5) micro-cracks, arising as a result of the intense thermomechanical impact, and (6) grooves, formed because of slipping of "passive" diamond grains over the ceramic surface.

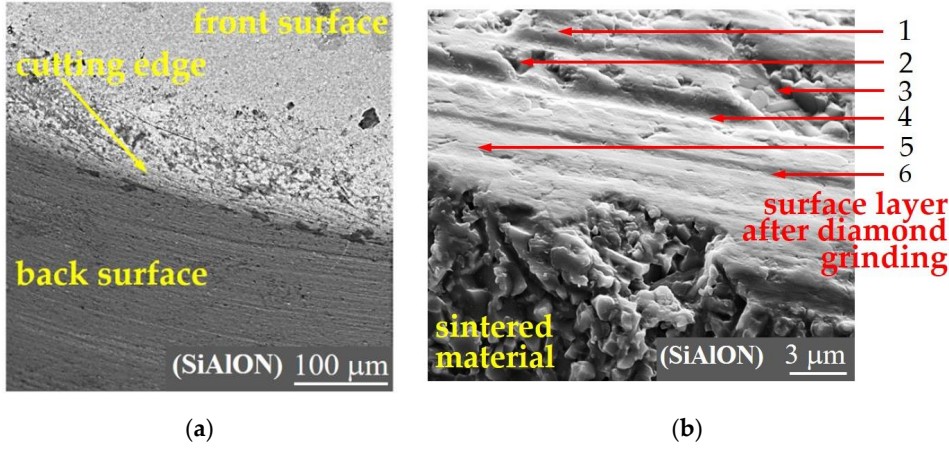

(**a**)    (**b**)

**Figure 5.** SEM-images of the general view of the cutting part of a round SiAlON ceramic insert (**a**) and the microstructure of the fracture boundary of sintered ceramics and the surface layer after diamond grinding (**b**): 1, micro ridges; 2, ripped-out single grains; 3, ripped-out conglomerate grains; 4, deep grooves; 5, micro cracks; 6, grooves.

The obtained 3D profilograms and SEM images of the microstructure of the surface layer of SiAlON ceramic inserts of the two studied groups are shown in Figure 6. The characteristic defects of the surface layer of industrially produced inserts (group I) with the typical topography [70] are noted in Figure 5 and visible on the profilograms (Figure 6a). The experimental data shown in Figure 6b demonstrate the pronounced positive changes occurring in the surface layer of industrially produced CCI due to additional diamond lapping and polishing (group II). Their use minimizes the defects formed during diamond grinding of the ceramic inserts (the maximum $R_t$ value was 0.37 μm compared to 3.6 μm for group I inserts) and improves the surface quality (the $R_a$ value was 0.014–0.026 μm compared to 0.28–0.3 μm for group I inserts).

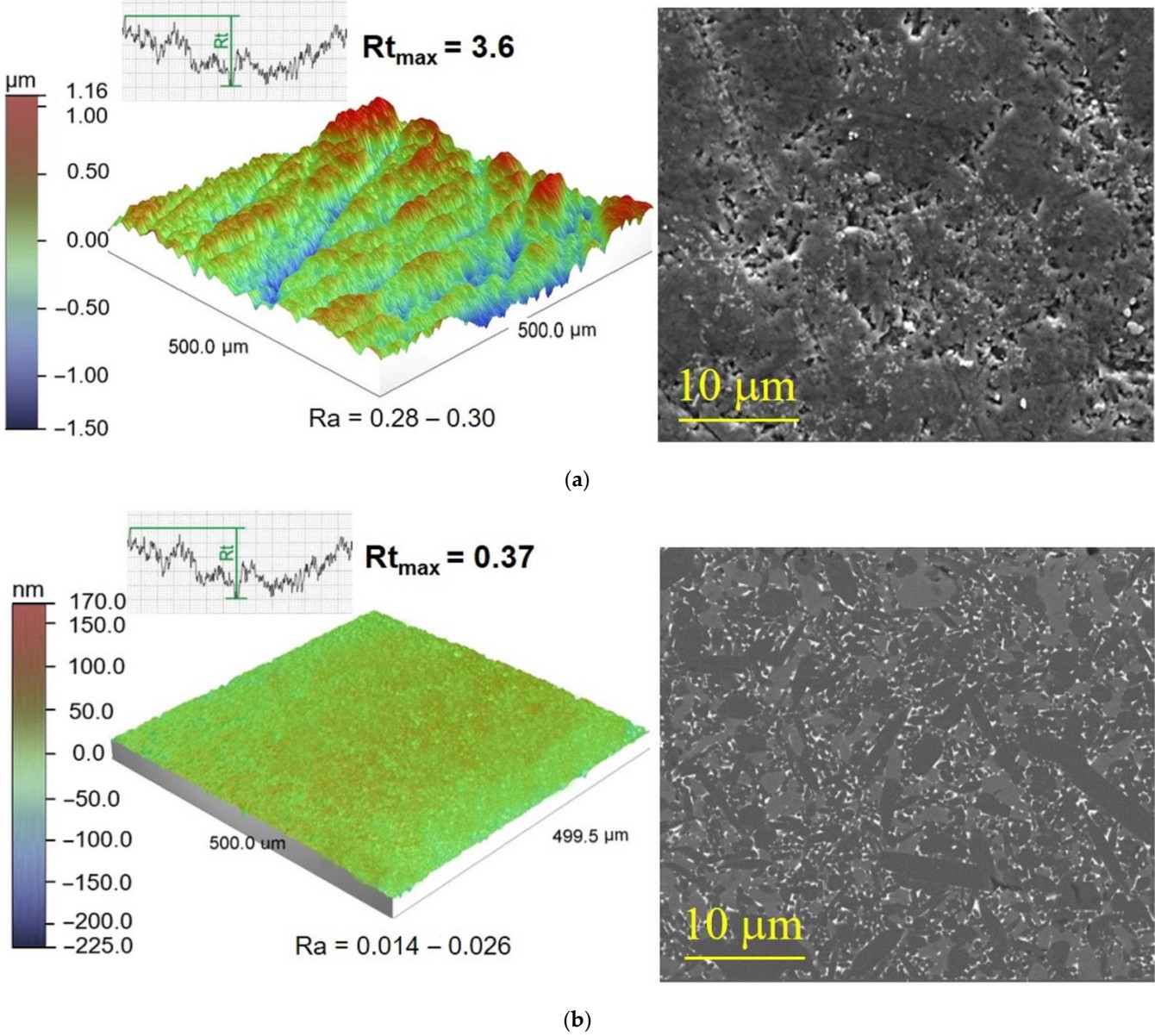

**Figure 6.** 3D profilograms (left) and SEM images (right) of the microstructure of the surface layer of SiAlON ceramic inserts after diamond grinding (**a**), and after diamond grinding, lapping, and polishing (**b**).

Table 3 summarizes data on the average values of the SiAlON CCI surface layer characteristics after various diamond abrasive machining methods. It can be seen that the

average value of $R_t$ after diamond grinding, lapping, and polishing was reduced many times—from 3.1 to 0.29 μm—while the parameter $R_a$ was reduced from 0.29 to 0.019 μm.

**Table 3.** Characteristics of the SiAlON ceramic inserts' surface layer after various diamond abrasive machining methods.

| No. | Diamond Abrasive Machining | Average Values of the Surface Layer Characteristics (According to the Measurement Result of 10 Samples) | | | |
|---|---|---|---|---|---|
| | | Crack Resistance $K_c$, MPa·m$^{1/2}$ | Microhardness HV, GPa | Roughness $R_a$, μm | Defect Layer Depth $R_t$, μm |
| 1 | Diamond grinding (group I) | 4.91 ± 0.35 | 15.99 ± 0.05 | 0.29 ± 0.025 | 3.1 ± 0.031 |
| 2 | Diamond grinding, lapping, and polishing (group II) | 5.26 ± 0.27 | 16.09 ± 0.05 | 0.019 ± 0.002 | 0.29 ± 0.032 |

The measurement results obtained by indentation crack resistance ($K_c$) of the SiAlON ceramic inserts' surface layer after various diamond abrasive machining methods (Table 3) revealed that the estimated indicator demonstrates a certain dependence on the defectiveness of the surface layer. The average $K_c$ value was increased by 7%—from 4.91 to 5.26 MPa·m$^{1/2}$—due to additional diamond lapping and polishing of the inserts (group II) compared to industrially produced CCI (group I). There was no noticeable effect on the microhardness of the condition of the CCI surface layer (Table 3).

Table 4 shows the experimentally obtained data of measured loads on the indenter corresponding to the chipping of the SiAlON ceramic inserts' cutting edges after various diamond abrasive machining methods (chipping and corresponding load were estimated based on the AE signal spectra). It has been found that additional lapping operations combined with polishing have a significant effect on the average value of the destructive load and the value of the spread of this indicator. For SiAlON CCI, an increase in the destructive load by ~30% was recorded.

**Table 4.** Loads on the indenter corresponding to the chipping of the SiAlON ceramic inserts' cutting edges after various diamond abrasive machining methods (measuring error of ±1.5 N).

| No. | Destructive Load, N | Diamond Abrasive Machining Methods | |
|---|---|---|---|
| | | Diamond Grinding (Group I) | Diamond Grinding, Lapping, and Polishing (Group II) |
| 1 | Average value | 28.9 | 37.6 |
| 2 | Max value | 33 | 40 |
| 3 | Min value | 26 | 36 |

Figure 7 shows the results of a study of the abrasion resistance of a SiAlON CCI surface layer formed by various diamond machining methods under the abrasive action of a rotating sphere. The presented dependences of the volume of worn material on the test time show that the presence of diamond grinding defects in the SiAlON ceramic inserts' surface layer (I group) significantly reduces the ability of the ceramics to resist abrasive wear. The ceramic inserts' lapping and polishing (group II) provide the lowest defectiveness of the surface layer and a decrease in the volume wear of the ceramic samples by ~1.8 times. It can be assumed that the so-called "edge effect" contributes to the decrease in the intensity of abrasion [71], considering that the microhardness of the SiAlON CCI surface layer of the two studied groups differs insignificantly (Table 3). The contact pads of the samples with a minimum number of defects restrain the development of a wear hole formed from the mechanical and abrasive effects of a rotating sphere. The boundaries of the surface layer containing multiple defects, shown in Figure 5, have a reduced ability to resist microfracture when exposed to an external load, and, as a result, the wear hole grows faster.

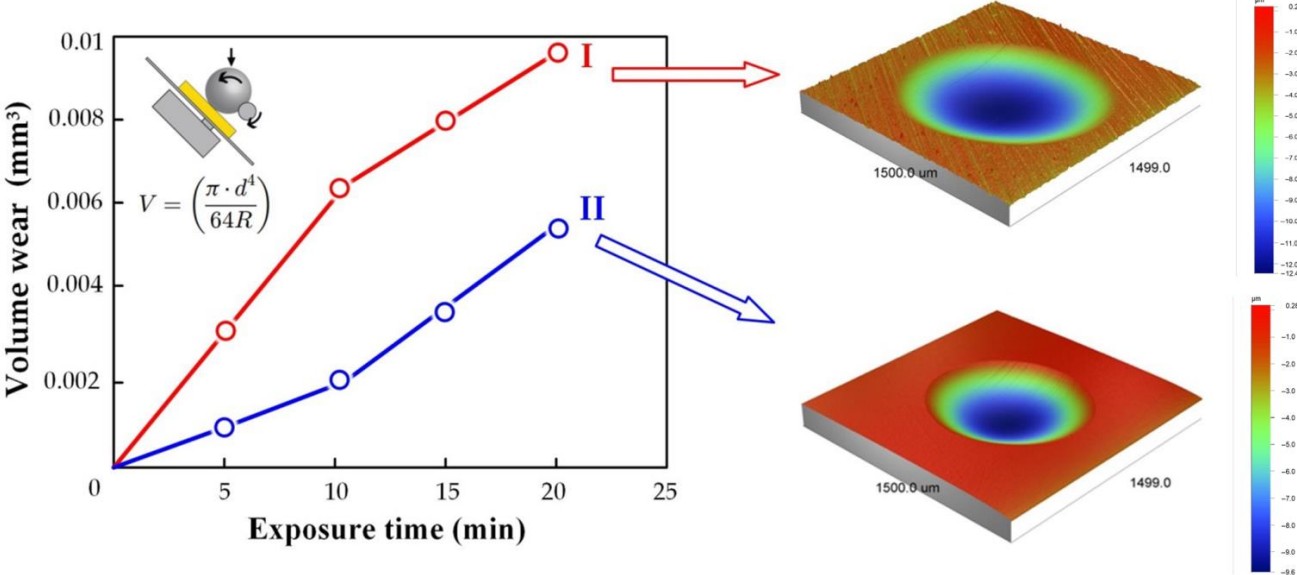

**Figure 7.** Dependences of the volume wear of SiAlON ceramic inserts with the different conditions of the surface layer after various diamond abrasive machining methods on the time of exposure to a rotating sphere and abrasive particles during testing, where (I) is diamond grinding, (II) is diamond grinding, lapping, and polishing.

Figure 8 shows the dependences of the coefficient of friction (COF) during high-temperature heating of SiAlON CCI after various abrasive machining methods such as diamond grinding (I) and additional lapping and polishing (II). The characteristic curves of the coefficient of friction changing over time demonstrate that minimizing the level of defects in the surface layer of samples (group II) provides an average of ~20% reduction in COF relative to samples with numerous defects (group I). Another feature that attracts attention is that at the stage of the run-in of contact surfaces in group I samples with high defectiveness, COF changes nonmonotonically, which may indicate the intensive adhesive setting of the SiAlON CCI surface layer with a counter body. For the group II samples, the change in COF is uniform throughout the entire test distance, which indicates more favorable conditions for frictional interaction. The observed patterns can be explained by a significant difference in the surface layer roughness achieved by the selected diamond abrasive machining methods.

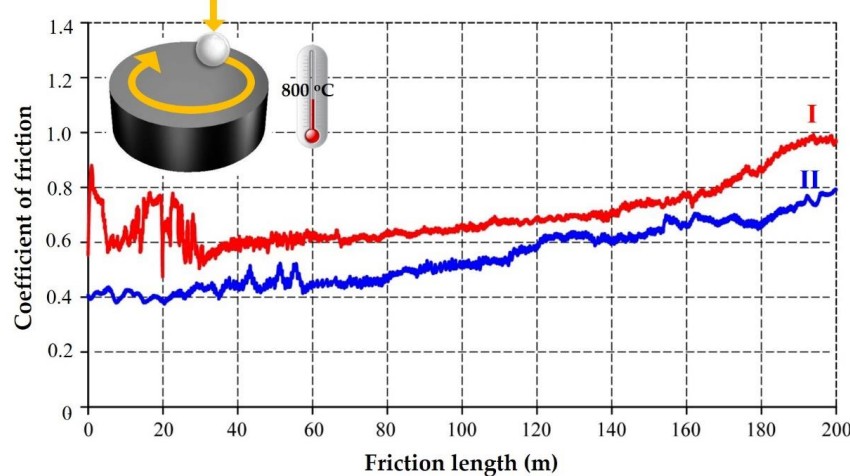

**Figure 8.** Dependences of the coefficient of friction during high-temperature heating of SiAlON ceramic inserts with different surface layer conditions on the length of the friction path, where (I) is diamond grinding, (II) is diamond grinding, lapping, and polishing.

*3.2. The Influence of Various Diamond Abrasive Machining Methods on the Durability of SiAlON Ceramic Inserts in Nickel-Based Superalloy Turning*

Figure 9a,b shows groups of "backface wear–cutting time" curves, constructed according to the results of laboratory tests of 10 faces of industrially produced round cutting inserts made of SiAlON when turning XH45MBTJuBP alloy at the cutting mode mentioned above. Experimental data illustrate (Figure 9a) the disadvantages associated with the CCI abovementioned low operational stability, which limits their industrial use. It can be seen that the curves of wear development over the cutting time for cutting inserts after diamond grinding (with the maximum number of surface defects) have a pronounced fan-shaped character with different wear rates of the cutting faces at the stage of run-in and regular wear. As a result, there is a considerable variation in the resistance until the failure criterion is reached. The authors of this work observed a similar pattern of wear development of ceramic inserts after diamond grinding when studying the turning of hardened steel 100CrMn6 using CCI made of $Al_2O_3$ + TiC ceramics [28].

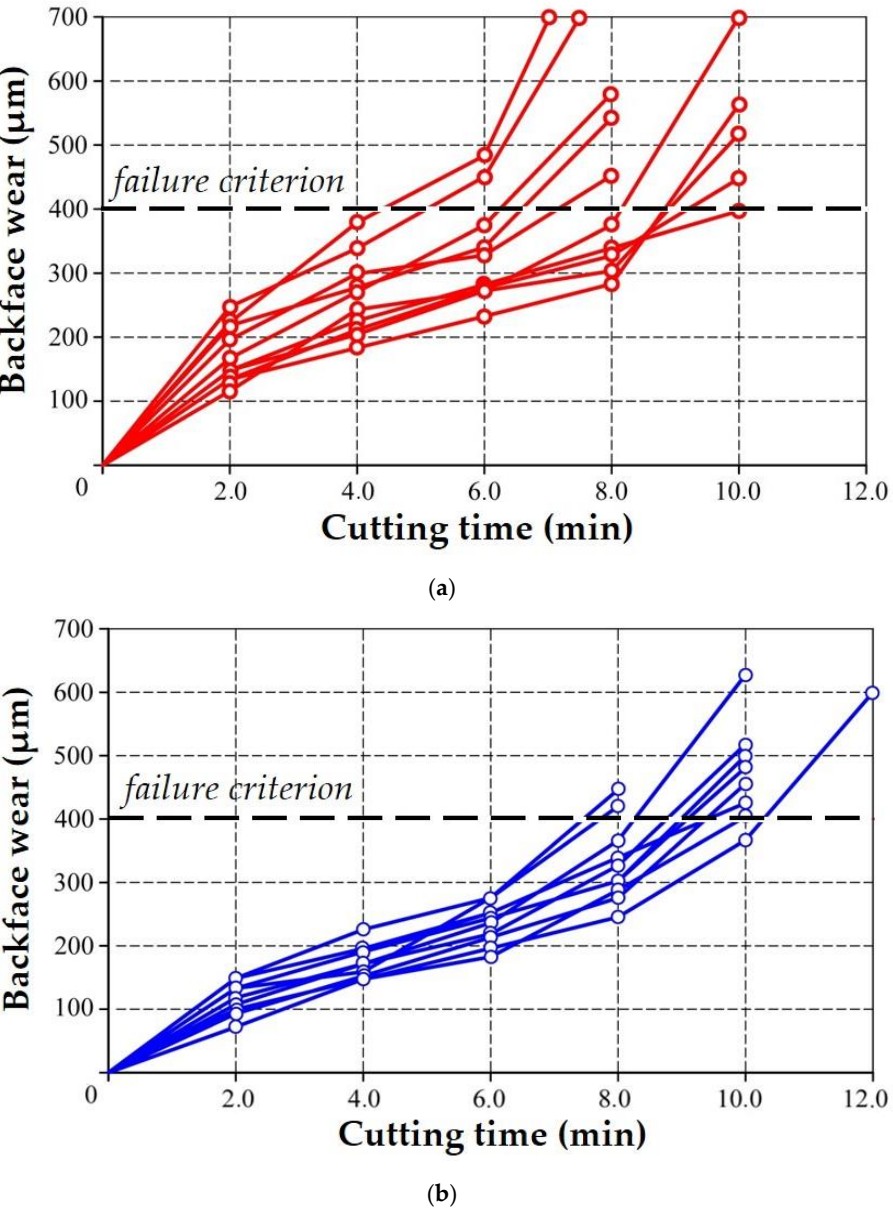

(a)

(b)

**Figure 9.** Group of "backface wear–cutting time" curves of the cutting faces of SiAlON ceramic inserts after diamond grinding (**a**), and diamond grinding, lapping, and polishing (**b**) when turning XH45MBTJuBP heat-resistant nickel alloy (V = 300 m/min, S = 0.2 mm/rev, t = 0.8 mm).

It should be noted once again that the surface layer of industrially produced inserts is replete with numerous defects that are stress concentrators, and when external loads are applied, the wear pattern is difficult to predict. Thus, the destruction of the cutting part can occur at any time (Figure 9a), while it can resemble the tearing out of entire conglomerates (Figure 1, option 3) When minimizing defects, i.e., repeated reduction of stress concentrators, the probability of destruction of the cutting part occurs mainly as a result of the gradual abrasion of the grains of the ceramic (Figure 1, option 1).

The additional lapping and polishing (with a minimum number of surface defects) have a significant impact on the nature of the development of cutting-edge wear over time (Figure 9b): the "backface wear–cutting time" curves have the form of an intertwining bunch of curves with significantly less variation in resistance values, while the ceramic inserts show higher operational stability.

It should be noted that the proposed research approach is unique (there are no other works related to the improving surface layer of SiAlON inserts for deposition of the unique coating structure for turning nickel-based alloys). The standard operating characteristic of a tool, the average tool life used by researchers, is an uninformative indicator for cutting ceramics. It can be seen (Figure 9a) that for ceramic inserts with a defective surface layer, the curves of development of wear over time along the back surface have a pronounced fan-shaped nature of the realizations of a random variable. Such a pattern of wear development over time is difficult to predict, and the operating time, until the accepted failure criterion is reached (in our case, 400 µm), has a large spread of values, which does not provide high operational reliability. Moreover, the approach proposed by the authors to reduce the defectiveness of the surface layer demonstrates a favorable effect on tool reliability, which is a key problem that hinders the use of ceramics in industry.

At the same time, if we compare the average values of wear on the flank surface of SiAlON CCI after diamond grinding (group I) and diamond grinding, lapping, and polishing (group II) on the turning time of the XH45MBTJuBP heat-resistant nickel alloy (Figure 10), the final difference in the average resistance of inserts with minimum and maximum defectiveness indices (according to the results of tests of 10 faces) is no more than 30% (9 min and 7 min, respectively). Therefore, the main effect of minimizing surface layer defects should be considered to be the reduction in the resistance spread, which provides more stable (predictable) operating conditions.

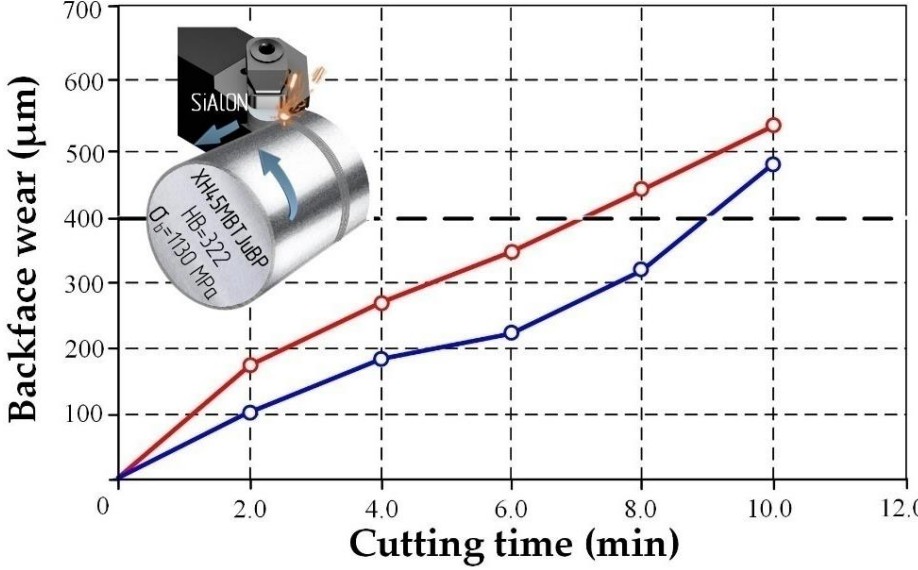

**Figure 10.** Dependences of the average backface wear value of SiAlON ceramic inserts after diamond grinding (red curve) and diamond grinding, lapping, and polishing (blue curve) on the cutting time when turning XH45MBTJuBP heat-resistant nickel alloy (V = 300 m/min, S = 0.2 mm/rev, t = 0.8 mm).

*3.3. Influence of the Condition of the Surface Layer of SiAlON Ceramic Inserts on the Quality of Formed (CrAlSi)N/DLC Coatings and Their Wear Resistance in Nickel-Based Superalloy Turning*

Figure 11 shows an image of the microstructure of a two-layer (CrAlSi)N/DLC coating formed on a SiAlON ceramic sample, the sublayer of which (CrAlSi)N has a columnar structure traditional for vacuum-arc nitride coatings. The outer DLC layer is characterized by an amorphous structure. The measurements showed that the microhardness of the formed outer DLC layer is 28 ± 2 GPa, and the average coefficient of friction during high-temperature heating (up to 800 °C) is ~0.4.

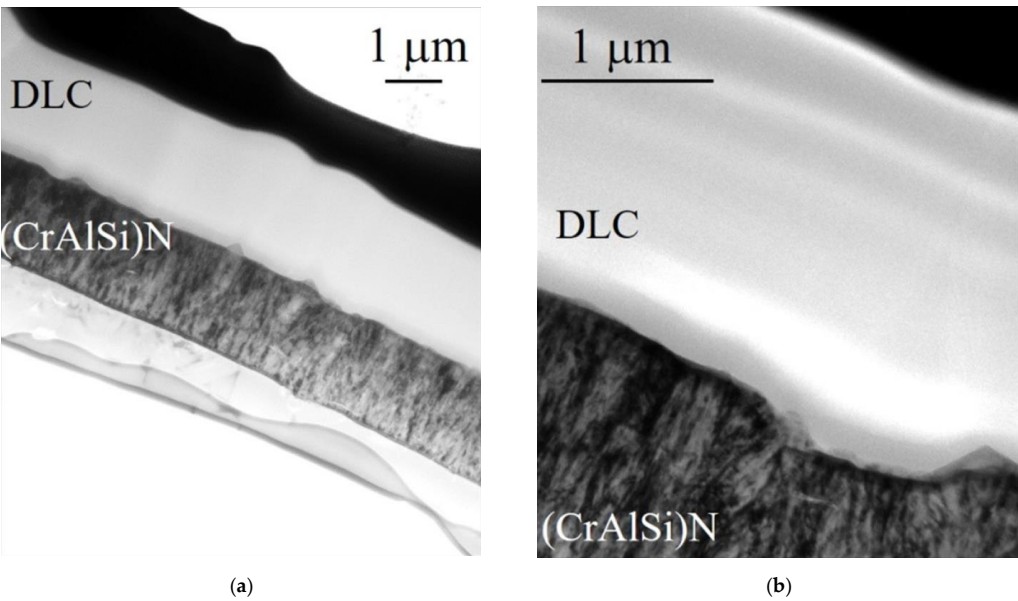

(**a**)          (**b**)

**Figure 11.** Microstructure of (CrAlSi)N/DLC coating formed on SiAlON ceramic inserts: (**a**) general view; (**b**) layer and sublayer.

Figure 12 shows images of the surface microstructure and profilograms of surface samples with (CrAlSi)N/DLC coatings deposited on SiAlON ceramic inserts with different states of the surface layer (groups I and II). It can be seen that the morphology of the deposited coatings significantly depends on the degree of defectiveness of the SiAlON CCI surface layer [72]. A thin-film coating cannot "heal" numerous defects that abound in industrially produced SiAlON inserts subjected to diamond grinding (the defects are listed in Figure 5) [73,74]. The coating only partly fills in rough surface defects—an experimental assessment of the surface layer of the samples by the $R_t$ parameter made it possible to establish that (CrAlSi)N/DLC coatings can reduce the depth of the defect layer by no more than 35–40%. Comparison of the data in Figure 12b allows us to conclude that the presence of various defects on the ceramic substrate during the deposition of thin-film coatings contributes to the formation of defects in their growth in the form of porosity, discontinuities, and deformation of crystallites (Figure 12a). With a minimum number of defects in the surface layer of ceramic samples, the deposited coatings are characterized only by morphological features characteristic of the PACVD process of DLC layer deposition (Figure 12b). The coatings do not contain visible pores and discontinuities, and the microstructure is represented by rounded crystallites [75,76].

The studies also found significant differences in the strength of the adhesive bond of (CrAlSi)N/DLC coatings depending on the condition (defectiveness) of CCI from SiAlON. It was found that coatings formed on defective substrates subjected to diamond grinding began to peel off at loads of ~33 N, and coatings deposited on samples after lapping and polishing began to peel off at loads of 42–44 N. This can be explained by the high density of micro defects in the SiAlON CCI surface layer leading to the formation of a large amount of porosity near the "coating-substrate" interface and an increase in internal stresses in the coatings [28].

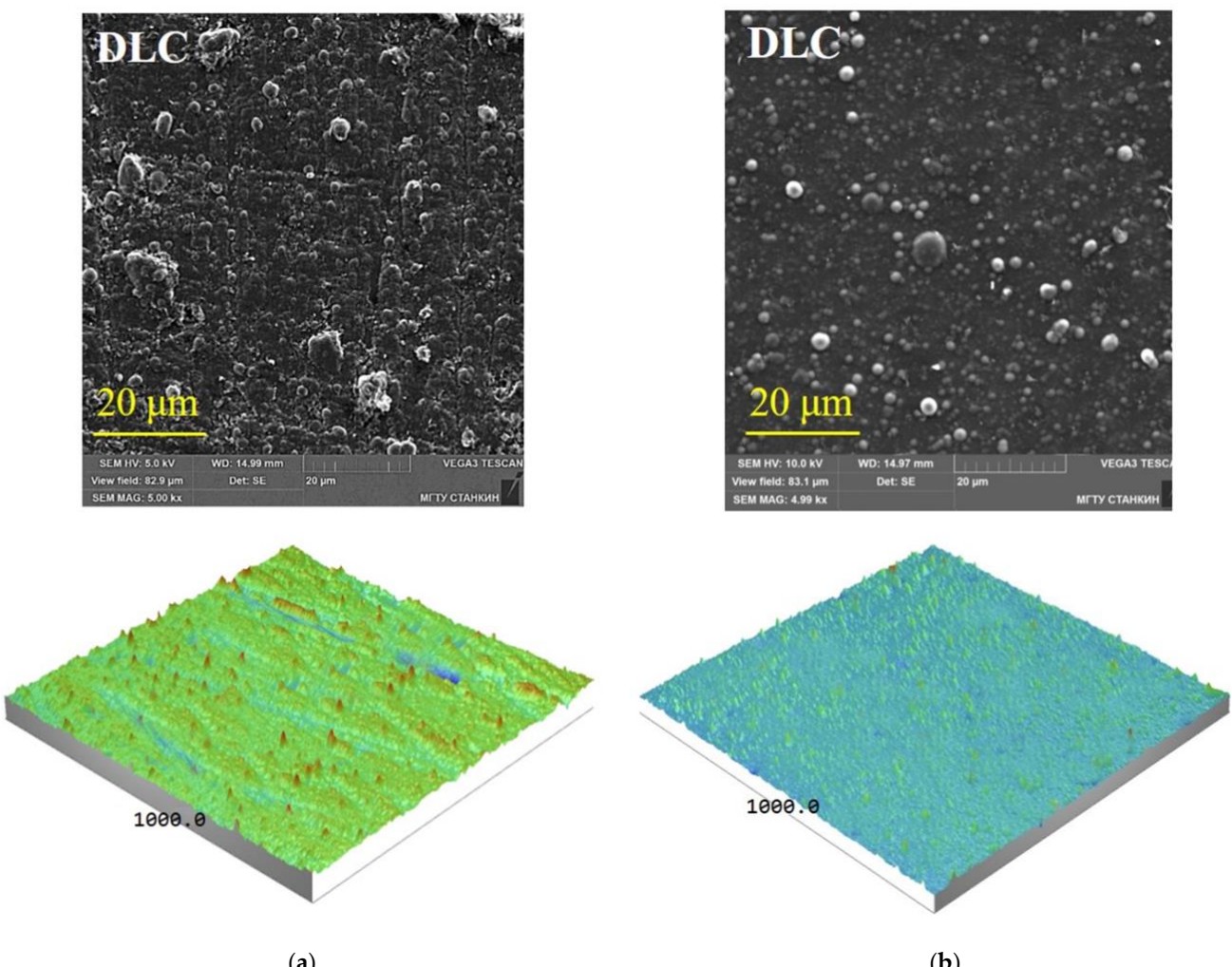

**Figure 12.** SEM images of the microstructure (top) and 3D profilograms (bottom) of (CrAlSi)N/DLC coatings deposited on SiAlON ceramic inserts with different surface layer conditions: (**a**) after diamond grinding; (**b**) after diamond grinding, lapping, and polishing.

The dependences of the volume of worn material on the test time presented in Figure 13, when compared with the data in Figure 7, allow us to conclude that the (CrAlSi)N/DLC coatings formed on ceramic inserts with different surface layer conditions are highly resistant to abrasion [77]. Compared with industrially produced SiAlON CCI, the formation of (CrAlSi)N/DLC coatings significantly reduces the volume wear of SiAlON ceramics when the surface layer is exposed to a rotating sphere and abrasive particles. A significant effect was found in the case of deposition of (CrAlSi)N/DLC coatings on samples of group I with a large number of defects (reduction of abrasive wear by 2.1 times) and their formation on samples of group II with a minimum number of defects (reduction of abrasive wear by 4.7 times). It can be assumed that the effective resistance to abrasive wear is provided by the high microhardness and low coefficient of friction of the SiAlON CCI contact surfaces achieved after (CrAlSi)N/DLC coating. The maximum effect found for SiAlON samples after diamond grinding, lapping, polishing, and (CrAlSi)N/DLC coating is achieved through a combination of high microhardness, low coefficient of friction, and high adhesion strength of the coatings.

It is necessary to perform cutting resistance tests for a more profound and comprehensive assessment of the influence of the surface layer condition of SiAlON CCI with deposed (CrAlSi)N/DLC coatings on the wear resistance of tool contact pads when interacting

with high-temperature nickel superalloys under conditions of increased mechanical and thermal loads.

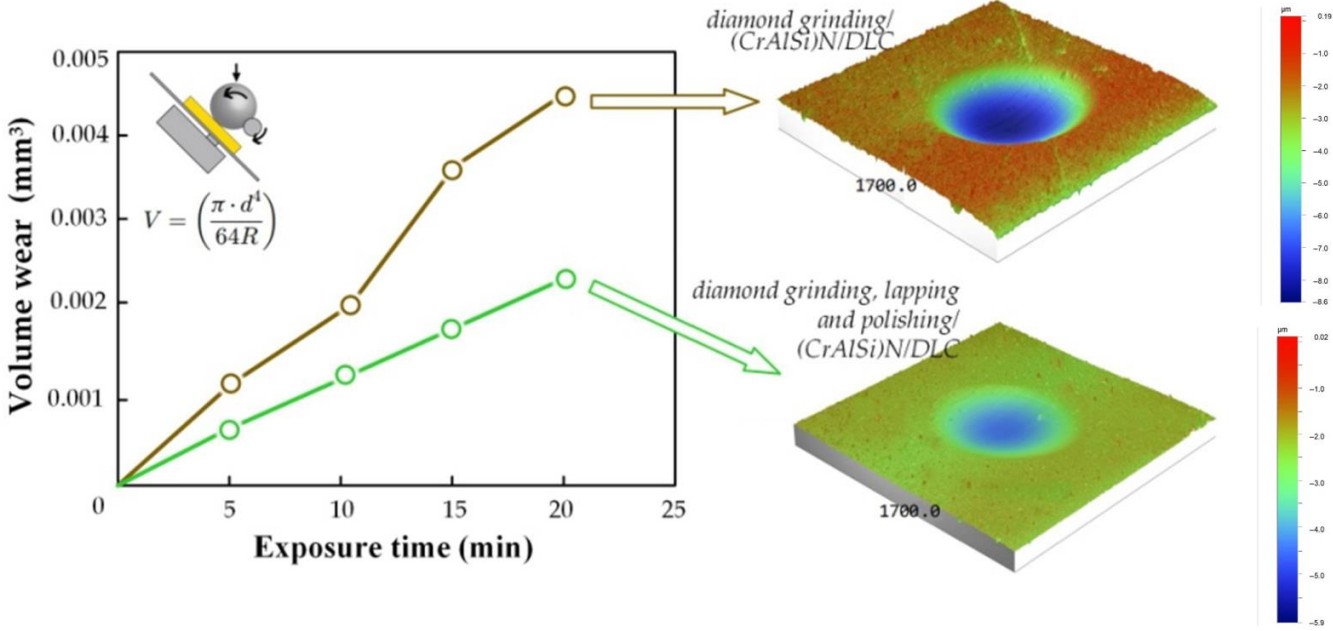

**Figure 13.** Dependences of the volume wear of SiAlON ceramic inserts with (CrAlSi)N/DLC coatings deposited on samples after various diamond abrasive machining methods on the abrasive exposure time.

Figure 14 shows experimentally obtained dependences of wear on the back surface of SiAlON ceramic inserts during turning of XH45MBTJuBP high-temperature nickel alloy, according to the test results of round CCI of three types:

- SiAlON after diamond grinding (industrially produced CCI), red curve;
- SiAlON after diamond grinding and (CrAlSi)N/DLC coating, brown curve;
- SiAlON after diamond grinding, lapping, polishing, and (CrAlSi)N/DLC coating, green curve.

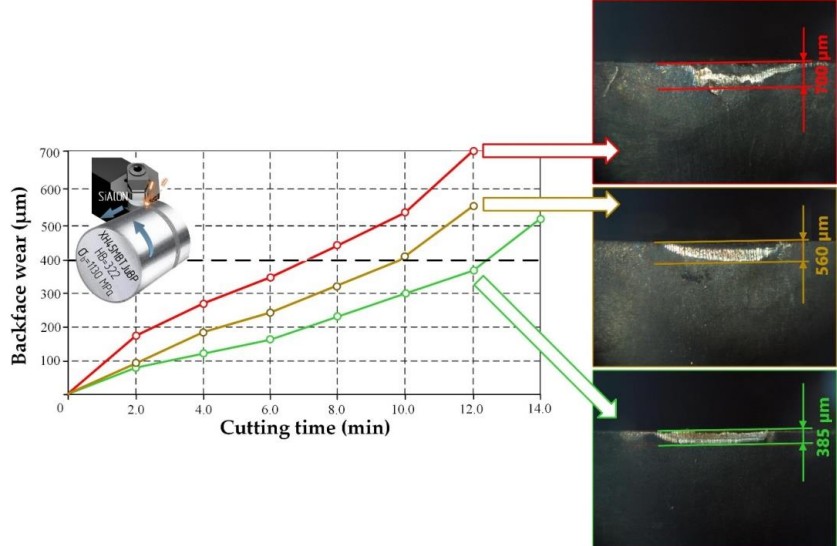

**Figure 14.** Dependences of the average backface wear of SiAlON ceramic inserts after diamond grinding (red curve), diamond grinding and (CrAlSi)N/DLC coating (brown curve), and diamond grinding, lapping, polishing, and (CrAlSi)N/DLC coating (green curve) on the cutting time when turning XH45MBTJuBP heat-resistant nickel alloy (V = 300 m/min, S = 0.2 mm/rpm, t = 0.8 mm).

The experimental results showed that the samples of SiAlON after diamond grinding, lapping, and polishing with (CrAlSi)N/DLC coating demonstrated the highest resistance to the accepted failure criterion (400 μm). The average durability was 12.5 min. At the same time, it was 9.9 min for the samples with (CrAlSi)N/DLC coatings formed on defective ceramic inserts and only 7 min for uncoated samples.

## 4. Conclusions

The study aimed to solve the actual scientific and technical problem of high accident brittleness and relatively short operational life of SiAlON cutting ceramics by changing the surface layer condition and double-layer coating of the (CrAlSi)N/DLC structure in the condition of extreme mechanical and thermal loads. The unique complex approach includes improving the surface of the ceramic inserts by additional diamond riding, lapping, and polishing when the tri-nitride sublayer improves the DLC coating adhesion in the conditions of chrome–nickel (Inconel 718 type) alloy machining.

The practical effect of diamond grinding and polishing in combination with the advanced double-layer (CrAlSi)N/DLC coating effect on the performance of industrially produced SiAlON ceramic cutting inserts in a chrome–nickel alloy (Inconel 718 type) turning was proven, and can be recommended for industrial application in production of responsible aircraft gas turbine engine parts.

The following conclusions were drawn:

1. The surface layer of industrially produced SiAlON ceramic inserts after diamond grinding combines numerous defects such as deep grooves, micro-cracks, ripped-out single grains, and conglomerates of grains of the sintered ceramic. The presence of a defective layer significantly reduces the resistance of the ceramic inserts' edges to chipping under external mechanical loads and also reduces the contact pads' ability to resist abrasive wear.

2. The conducted studies allowed us to obtain data proving the strong influence of the condition of the surface layer (presence of defects) of SiAlON ceramic inserts on their operational stability (resistance spread) when turning heat-resistant nickel superalloy under conditions of increased cutting speeds and cross-section of the cut layer. At the same time, using various diamond abrasive machining methods, in particular, additional lapping and polishing, allows for minimization of the defective layer formed during diamond grinding and reduction of the resistance spread.

3. Deposition of thin-film two-layer (CrAlSi)N/DLC coatings on the surface of industrially produced SiAlON ceramic inserts significantly improves the characteristics of tool ceramics, such as the microhardness of the surface layer increasing, and the coefficient of friction decreasing during high-temperature heating while the abrasion resistance also increases. However, (CrAlSi)N/DLC coatings are not able to "heal" numerous defects of the surface layer but can only reduce the depth of the defective layer. The defects in the surface layer of SiAlON ceramic inserts contribute to forming porous and discontinuous coatings with low adhesive bond strength.

4. The maximum effect when turning heat-resistant nickel superalloys under increased cutting speed and cross-section of the cut layer is achieved by using a combined surface treatment method, where lapping and polishing minimize the diamond grinding defects and the subsequent (CrAlSi)N/DLC coating provides an increase in the microhardness of the surface layer and a decrease in the coefficient of friction on the contact surfaces. The developed approach makes it possible to increase the resistance of SiAlON ceramic inserts by a factor of 1.78 compared to industrially produced inserts.

**Author Contributions:** Conceptualization, S.N.G.; methodology, M.A.V.; software, M.A.V. and A.A.O.; validation, M.A.V.; formal analysis M.A.V.; investigation, M.A.V.; resources, S.N.G.; data curation, A.A.O.; writing—original draft preparation, M.A.V.; writing—review and editing, M.A.V.; visualization, M.A.V. and A.A.O.; supervision, S.N.G.; project administration, M.A.V.; funding acquisition, S.N.G. All authors have read and agreed to the published version of the manuscript.

**Funding:** This work was supported financially by the Ministry of Science and Higher Education of the Russian Federation (project No. FSFS-2021-0006).

**Data Availability Statement:** Data are available in a publicly accessible repository.

**Acknowledgments:** The study was carried out on the equipment of the Center of collective use of MSUT "STANKIN" supported by the Ministry of Higher Education of the Russian Federation (project No. 075-15-2021-695 from 26 July 2021, unique identifier RF 2296.61321X0013).

**Conflicts of Interest:** The authors declare no conflict of interest.

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
