# Peer review of "Investigation of Surface Layer Condition of SiAlON Ceramic Inserts and Its Influence on Tool Durability When Turning Nickel-Based Superalloy"

_technologies, doi:10.3390/technologies11010011_

Round 1
Reviewer 1 Report
Suggestions for improving the manuscript are as follows:
1. The abstract should be rewritten. The abstract must be presented in a clear way: problematic, objective, idea, methods, results, quantitative comparison of results with significant findings, conclusions.
2. In the last paragraph of the Introduction section, write explicitly the scientific contribution of your research and scientific hypothesis.
3. How was the cutting tool selected?
4. More geometry data needs to be displayed for the cutting tool (not only diameter and thickness).
5. How are the cutting conditions (cutting speed, feed, and depth of cut) selected?
6. Which tools were used during machining?
7. How are the selected values of technological modes shown in table 2? Why exactly those values?
8. Estimate the measurement uncertainty of the obtained results.
9. Compare the obtained results with the results of previous research.
10. The conclusions do not need to repeat the results. The Conclusion section should be rewritten. Highlight your scientific contribution. Highlight the benefits of your research. Define shortcomings and future research.
Author Response
Response to Reviewer 1 Comments
Dear reviewer,
Thank you so much for your kind evaluation of our work. We do agree with all your proposals and comments and have modified the manuscript according to them.
We hope that the manuscript will be suitable for publishing in Technologies and will attract many potential readers of the journal with your comments. The introduced corrections in the text of the manuscript are marked yellow.
Kind regards,
Authors.
Reviewer comments
Point 1: The abstract should be rewritten. The abstract must be presented in a clear way: problematic, objective, idea, methods, results, quantitative comparison of results with significant findings, conclusions.
Response 1: Thank you for pointing it out. The abstract was revised.
Point 2: In the last paragraph of the Introduction section, write explicitly the scientific contribution of your research and scientific hypothesis.
Response 2: Thank you for the remark. The last paragraph was revised, and we hope it looks better in its current form.
Point 3: How was the cutting tool selected?
Response 3: Thank you, we agree that it was not clearly explained. As a tool, which is the object of research, the least studied were chosen (the focus of researchers until now has been mainly square and rhombic plates made of Al2O3 + TiC ceramics, less often Si3N4), but at the same time, ceramic round plates made of SiAlON, designed for processing parts of gas turbine engines from heat-resistant chromium-nickel alloys (along with titanium alloys, they are the main structural material for the manufacture of aircraft engine elements).
Point 4: More geometry data needs to be displayed for the cutting tool (not only diameter and thickness).
Response 4: Thank you for this remark. Actually, it is only geometry provided at the cutting ceramic (oxide-nitride) insert http://www.imc-companies.com/TaeguTec/ttkCatalog/Family.aspx?fnum=150&mapp=IS&app=26&GFSTYP=M&fr=1 (it can be seen not only at the catalog but in reality). We have already considered similar tool in our work made of another cutting ceramic (nitride) https://www.mdpi.com/2079-6412/12/4/469. Additional angle appeared only at the moment of insert placement in the holder (the clearance angle of 5°, the rake angle of -5° (negative!), and the lead angle of 46° was set when placed in the holder, added to the text). It should be noted that cutting ceramic is a material that can be called difficult to machine due to its high hardness and brittleness; only simple shapes can be obtained. However, we also worked in the field of shaping cutting ceramics for many years and even achieved some results in it: https://www.mdpi.com/2227-7080/11/1/6 (surface micro texturing). SiAlON can be considered one of the most inconvenient materials for coating and micro texturing/machining but exhibits superior wear resistance compared to carbide tools, especially under extreme load conditions (semi-finishing in the temperature range up to 800°C with increased depth of cut and feed).
Point 5: How are the cutting conditions (cutting speed, feed, and depth of cut) selected?
Response 5: Thank you for pointing it out. Increased values have been selected from the practice of semi-finishing difficult-to-machine alloys based on nickel and titanium. Increased feed at a simultaneously high cutting speed provides more intense heat and power loads on the tool when the probability of premature failure of the ceramic insert increases sharply, and it is under these conditions that the role of the surface layer increases. The relevant passage is added.
Point 6: Which tools were used during machining?
Response 6: Thank you; all used cutting and measuring tools are mentioned in Section 2. For machining (turning), it is cutting insert and holder mentioned above:
http://www.imc-companies.com/TaeguTec/ttkCatalog/Family.aspx?fnum=150&mapp=IS&app=26&GFSTYP=M&fr=1 (insert)
https://fliphtml5.com/kqrw/ogfz/basic/1501-1543 (insert, page 1518)
https://www.sandvik.coromant.com/ru-ru/product-details?c=CRSNR%203232P%2019-ID&tab=MatchingADINTMS (holder)
It is revised in the text.
Point 7: How are the selected values of technological modes shown in table 2? Why exactly those values?
Response 7: Thank you; the modes mentioned in Table 2 were worked out in previous years and were chosen to ensure maximum adhesion of coatings [10,38,41]. It was revised in the text.
Point 8: Estimate the measurement uncertainty of the obtained results.
Response 8: Thank you; the relevant data are added. To minimize the error of experiments, all experiments were duplicated 10 times.
Point 9: Compare the obtained results with the results of previous research.
Response 9: Thank you, it should be noted that the proposed research approach is uniquely original (there is no other works related to the improving surface layer of SiAlON inserts for deposition of the unique coating structure for turning nickel-based alloys). The standard operating characteristic of a tool, the average tool life used by researchers, is an uninformative indicator for cutting ceramics. It can be seen (Figure 9, a) that for ceramic inserts with a defective surface layer, the curves of development of wear over time along the back surface have a pronounced fan-shaped nature of the realizations of a random variable. Such a pattern of wear development over time is difficult to predict, and the operating time, until the accepted failure criterion is reached (in our case, 400 µm), has a large spread of values, which does not provide high operational reliability. Moreover, the approach proposed by the authors to reduce the defectiveness of the surface layer demonstrates a favorable effect on tool reliability, which is a key problem that hinders the use of ceramics in industry. The relevant passage is added.
Point 10: The conclusions do not need to repeat the results. The Conclusion section should be rewritten. Highlight your scientific contribution. Highlight the benefits of your research. Define shortcomings and future research.
Response 10: Thank you for the question. Experimental studies have shown that the role of the surface layer state is key and largely determines the performance of coatings deposed to cutting ceramics, which are operated at increased feeds and high cutting speeds. In this regard, further developments and research are needed in the field of various methods of targeted action on the surface layer to minimize its defectiveness before coating, for example, using a laser, electron beam, jet, and other types of exposure. It is because multi-stage abrasive processing, such as additional finishing and polishing, is suitable for research purposes to prove the key influence of the role of the surface layer, but in practice, something more productive and cost-effective is needed. The conclusions are revised.
Reviewer 2 Report
Dear authors,
Thank you very much for submitting your article for evaluation to Technologies Journal. This article is very interesting but requires some correction according to the comments below:
- The state-of-the-art (introduction section) is short, you should add more description about research in this field, e.g. Zajac, J., et al (2020). Prediction of Cutting Material Durability by T= f (vc) Dependence for Turning Processes. Processes, 8(7), 789.; Panda, et al. (2013). The analysis of ceramic cutting tools durability in machining process of steel C60 applied according to standard ISO 3685. In Applied Mechanics and Materials; or Kaladhar, M., Subbaiah, K. V., & Rao, C. S. (2013). Optimization of surface roughness and tool flank wear in turning of AISI 304 austenitic stainless steel with CVD coated tool. Journal of Engineering Science and Technology, 8(2), 165-176.
- figure 1 – please add a reference
- table 2 – “technological modes” – check units °C and situated in square bracket
- table 3 – units should be situated in square brackets
- figure 7, 13 – the scale is missing
- conclusion – please add the contribution of your research to practice and science
Author Response
Response to Reviewer 2 Comments
Dear reviewer,
Thank you so much for your kind evaluation of our work. We agree with all your proposals and comments and have modified the manuscript accordingly.
We hope the manuscript will be suitable for publishing in Technologies and attract many potential journal readers with your comments. The introduced corrections in the text of the manuscript are marked green.
Kind regards,
Authors.
Reviewer comments
Point 1: The state-of-the-art (introduction section) is short, you should add more description about research in this field, e.g. Zajac, J., et al (2020). Prediction of Cutting Material Durability by T= f (vc) Dependence for Turning Processes. Processes, 8(7), 789.; Panda, et al. (2013). The analysis of ceramic cutting tools durability in machining process of steel C60 applied according to standard ISO 3685. In Applied Mechanics and Materials; or Kaladhar, M., Subbaiah, K. V., & Rao, C. S. (2013). Optimization of surface roughness and tool flank wear in turning of AISI 304 austenitic stainless steel with CVD coated tool. Journal of Engineering Science and Technology, 8(2), 165-176.
Response 1: Thank you so much for your kind recommendation. I have watched the mentioned publications and found them attractive for our future work. Unfortunately, we cannot mention them in the current paper due to etic issues since we suppose that some of these papers are of the reviewer. However, we have extended the introduction section.
Point 2: - figure 1 – please add a reference
Response 2: Thank you for asking it. The figure is original. The picture of a Sandvik holder with a cutting insert of TaeguTec was taken by a private smartphone. It should be noted that TaeguTec inserts are fully compatible with Sandvik holders as presented here TaeguTec holders http://www.imc-companies.com/TaeguTec/ttkCatalog/Family.aspx?fnum=10175&mapp=IS&app=26&GFSTYP=M.
Point 3: - table 2 – “technological modes” – check units °C and situated in square bracket
Response 3: Thank you for your kind suggestion. Table 2 was revised, and now it looks like a separate column that helps make the data more readable. It should be noted we are afraid to place all units in square brackets since it will reduce workability for editors when working with references. If the editor asks it as well, we will be happy to provide the units in square brackets.
Point 4: - table 3 – units should be situated in square brackets
Response 4: Thank you so much for your kind suggestion. We would like to keep it as it is at this stage to make it more understandable for editors where units and where references are, but if the editors ask to provide everything in square brackets, we will do it with pleasure.
Point 5: - figure 7, 13 – the scale is missing
Response 5: Thank you for noticing it; the figures are revised.
Point 6: - conclusion – please add the contribution of your research to practice and science.
Response 6: Thank you, the conclusions are revised.
Reviewer 3 Report
This paper study the influence of the surface layer condition of SiAlON ceramic cutting inserts processed by diamond grinding and polishing on the tool operability in high-speed turning of nickel alloy. This work is interesting and has been well presented. The following questions should be revised.
1. Introduction: the writing of the research purpose and current situation is fine. The unit of feeds should be revised in “with an increase in the cross-section of the cut layer (feeds of more than 0.15 rpm)”.
2. The units of tool size should be added in Fig. 2. Besides the units should be improved in “The measurements showed that the micro-hardness of the formed outer DLC layer is 28 ± 2 GPa, and the average coefficient of friction during high-temperature heating (up to 800 oC) is ~0.4.”
3. The wear chamfer value of 400 µm is very important in the following tool wear evaluation. Where is the failure criterion from? Some reference should be added here. Usually, the wear chamfer value of VB0.3 mm is widely used in turning according to ISO criterion.
4. Figure 6 and 12 show the 3D profilograms and SEM images of the microstructure of the surface. Similar representation methods are also used for machined surface, some reference should be added here. See a lately publication “Ironing effect on surface integrity and fatigue behavior during ultrasonic peening drilling of Ti-6Al-4V”.
5. Table 2 shows the range of electrical discharge machining factors. The authors should reconsider the necessity of describing EDM content in a large length in this section.
6. The quality of the language is ok. In general, the language could be refined carefully with the help of a native English speaker.
Author Response
Response to Reviewer 3 Comments
Dear reviewer,
Thank you so much for your kind evaluation of our work. We agree with all your proposals and comments and have modified the manuscript accordingly.
We hope the manuscript will be suitable for publishing in Technologies and attract many potential journal readers with your comments. The introduced corrections in the text of the manuscript are marked blue.
Kind regards,
Authors.
Reviewer comments
Point 1: Introduction: the writing of the research purpose and current situation is fine. The unit of feeds should be revised in “with an increase in the cross-section of the cut layer (feeds of more than 0.15 rpm)”.
Response 1: Thank you so much for pointing it out. It was revised.
Point 2: The units of tool size should be added in Fig. 2. Besides the units should be improved in “The measurements showed that the micro-hardness of the formed outer DLC layer is 28 ± 2 GPa, and the average coefficient of friction during high-temperature heating (up to 800 oC) is ~0.4.”
Response 2: Thank you for noticing it. Figure 2 and measuring units are revised.
Point 3: The wear chamfer value of 400 µm is very important in the following tool wear evaluation. Where is the failure criterion from? Some reference should be added here. Usually, the wear chamfer value of VB0.3 mm is widely used in turning according to ISO criterion.
Response 3: Thank you for your kind suggestion. It is true that most of the research group traditionally use values of flank wear of 0.3 mm:
- DOI 10.1016/0043-1648(88)90159-7
- DOI 10.1016/S0924-0136(01)00877-9
- DOI 10.1016/S0043-1648(01)00570-1
At the same time, we found some papers with variation of the criterion from 0.1 to 0.35:
- Kumar, R., Sahoo, A.K., Mishra, P.C., Das, R.K. An investigation to study the wear characteristics and comparative performance of cutting inserts during hard turning. International Journal of Machining and Machinability of Materials. 2018, 20(4), 320-344.
- Devin L.N.;Vilgelm M. Failure probability prediction of polycrystalline CBN cutting tools. Sverkhtverdye Materialy 1992, 6, 41 – 46.
Here is considered 2 criteria of 0.3 and 0.5 mm:
- https://link.springer.com/article/10.1007/s11148-018-0241-1
Here is even an article of 0.45 mm
- https://doi.org/10.3390/coatings8080287
Our research group has taken it of 0.4 mm in many published works:
- Grigoriev, S.N.; Volosova, M.A.; Fedorov, S.V.; Migranov, M.S.; Mosyanov, M.; Gusev, A.; Okunkova, A.A. The Effectiveness of Diamond-like Carbon a-C:H:Si Coatings in Increasing the Cutting Capability of Radius End Mills When Machining Heat-Resistant Nickel Alloys. Coatings 2022, 12, 206.
- Vereschaka, A.A., Grigoriev, S.N., Vereschaka, A.S., Popov, A.Y., Batako, A.D. Nano-scale multilayered composite coatings for cutting tools operating under heavy cutting conditions. Procedia CIRP 2014, 14, pp. 239-244.
- Uhlmann, E., Hühns, T., Richarz, S., Reimers, W., Grigoriev, S. Development and application of coated ceramic cutting tools. Industrial Ceramics 2009, 29(2), 113-118.
- https://www.mdpi.com/2079-6412/12/12/1801
- https://www.mdpi.com/2079-6412/12/4/469 (here is the similar picture of the cutting insert holder, Figure 2b)
When choosing the criterion of the flank wear, it should be noted that finishing 0.3 mm is an optimum value, but only for the tools that are subjected to regrinding. The ceramic insert is not related to this group of cutting tools. They are much more expensive than hard alloy tools and cannot be reused after achieving the criterion. In most cases, this type of cutting tool is used carefully with a relatively small depth of cut. At the same time, in this article, we use a relatively large depth of cut and feed to achieve mentioned purposes to maximize tool life that is not subject to regrinding.
Thus, we completely agree with the reviewer's remark that the recommended flank wear is 0.3 mm. It is a fair remark when ceramics are used in finishing. Then the surface roughness has a crucial importance. Indeed, an increase in flank wear greater than 0.3 mm leads to intense surface roughness degradation. In our case, the chosen cutting modes are not suitable for finishing but semi-finishing, after which finishing is foreseen. In this condition, the tool is foreseen to develop its resource to the maximum. Thus, flank wear of 0.4 mm is accepted. The authors of this paper have repeatedly considered the mentioned flank wear value in previously published papers mentioned above. Finally, with the use of additional surface technologies, it is possible to further increase the tool life even under increased cutting conditions.
The relevant passage is added.
Point 4: Figure 6 and 12 show the 3D profilograms and SEM images of the microstructure of the surface. Similar representation methods are also used for machined surface, some reference should be added here. See a lately publication “Ironing effect on surface integrity and fatigue behavior during ultrasonic peening drilling of Ti-6Al-4V”.
Response 4: Thank you so much for your kind suggestion. We found this article attractive for our further research. However, we cannot add it right away due to etic issues since it can be a publication of the reviewer. We added some other references. We hope to meet understanding.
Point 5: Table 2 shows the range of electrical discharge machining factors. The authors should reconsider the necessity of describing EDM content in a large length in this section.
Response 5: Thank you for noticing it; that was a technical mistake. The correct title of the table is processing factors. It was revised. However, if the reviewer finds it redundant information, we can always remove it from the text.
Point 6: The quality of the language is ok. In general, the language could be refined carefully with the help of a native English speaker.
Response 6: Thank you so much for your kind evaluation. The manuscript was checked once again by a native speaker. Some corrections are introduced.
Reviewer 4 Report
1. Please add the processing detials about diamond grinding, lapping, and polishing.
2. As a contast in Fig.5, please add the SEM-images of the general view of the cutting part of a round SiAlON ceramic insert by diamond grinding, lapping, and polishing.
Author Response
Response to Reviewer 4 Comments
Dear reviewer,
Thank you so much for your kind evaluation of our work. We agree with all your proposals and comments and have modified the manuscript according to them.
We hope that the manuscript will be suitable for publishing in Technologies and will attract many potential readers of the journal with your comments. The introduced corrections in the text of the manuscript are marked marine blue.
Kind regards,
Authors.
Reviewer comments
Point 1: Please add the processing detials about diamond grinding, lapping, and polishing.
Response 1: Thank you for pointing it out. Additional processing of industrial ceramic inserts was carried out on a Lapmaster Wolters finishing and polishing machine with special finishing and polishing wheels using various diamond suspensions (grit of 50/40, 40/28 for finishing and 10/7, 5/3 for polishing) at a cutting speed of 3 m/s. Finishing has taken of 28 min, and polishing was of 16 min. The relevant data is highlighted.
Point 2: As a contast in Fig.5, please add the SEM-images of the general view of the cutting part of a round SiAlON ceramic insert by diamond grinding, lapping, and polishing.
Response 2: Thank you for pointing it out. Unfortunately, we did only SEM images of the surface before it. We will take this remark into account for further research.
Reviewer 5 Report
The paper is good and and well presented. Following issues need to be addressed:
1. The authors should clarify what is meant by DLC in detail. How it differs from conventional diamond coating.
2. Details about lapping and polishing is required. How the parameters have been selected?
3. What could be the reasons for different wear behavior among group of curves (10 curves) in Figure 9(a) and (b).
4. Why round cutting inserts have been selected?
5. It would be better if results are presented for more than one cutting speed and the variations should be discussed for the three cases.
Author Response
Response to Reviewer 5 Comments
Dear reviewer,
Thank you so much for your kind evaluation of our work. We do agree with all your proposals and comments and have modified the manuscript according to them.
We hope that the manuscript will be suitable for publishing in Technologies and will attract many potential readers of the journal with your comments. The introduced corrections in the text of the manuscript are marked grey.
Kind regards,
Authors.
Reviewer comments
Point 1: The authors should clarify what is meant by DLC in detail. How it differs from conventional diamond coating.
Response 1: Thank you for pointing it out. The main difference is in providing a unique sublayer of (CrAlSi)N. The role of the sublayer is better explained in https://www.mdpi.com/2079-6412/11/5/532 [10], where similar multilayer coating is deposed to the end mill produced of SiAlON+TiN ceramics (our research group provided studies of the whole production cycle of the cutting tool manufacturing mainly to the purposes of producing the parts of gas turbine engines).
The deposition of a two-layer coating (CrAlSi)N/DLC significantly affects the properties of the surface and surface layer of cutting ceramics. The coating significantly modifies the relief of a thin surface layer and significantly affects the size and shape of surface microroughness and some defects formed at the stage of diamond sharpening. It somehow fills the micro-grooves on the surface, providing a kind of surface layer “healing.” For the (CrAlSi)N/DLC coating, the friction coefficient remains at a low level for a relatively long time and varies slightly within the range of 0.09–0.15, and begins to increase only at the end of the tests, reaching 0.72. It is important to note that the coating contributes to improving the quality of the surface layer of the workpiece being processed by changing the conditions of interaction with the cutting tool in the tribocontact zone.
Another article related to sublayer is https://www.mdpi.com/2079-6412/10/11/1038 [41]. The conclusions are as follows:
- When nanoindenting with a load of 4.0 mN, the positive effect of the properties of the substrate with a preformed sublayer (CrAlSi)N is sharply manifested; the index of plasticity (ratio of hardness and modulus of elasticity) for the (CrAlSi)N/DLC was almost 0.13, while for a single-layer DLC, this indicator was at the level of 0.1. The formation of the (CrAlSi)N intermediate sublayer has a significant effect on the ability of the DLC to resist elastoplastic deformations.
- Higher plasticity of (CrAlSi)N/DLCs and a less stressed state at the substrate-coating interface had an essential effect on increasing the adhesive bond strength of the coating to the substrate compared to a single-layer DLC. The coating separation from the sample with a single-layer DLC, which has lower plasticity (greater modulus of elasticity), occurs at a significantly (almost two times) lower load. Simultaneously, the wear rate under the influence of abrasive particles on (CrAlSi)N/DLC-coated hard alloy samples are 2.4 times lower than uncoated samples and 1.5 times lower than single-layer DLC-coated samples.
- When cutting, the single-layer DLC very quickly ceases to perform protective functions, and the contact conditions on the front and flank surfaces of the tool approach those that occur when using cutters without coatings. The deposition of (CrAlSi)N/DLC alone provided an increase in the durability of end mills up to two times. The role of the (CrAlSi)N sublayer in milling steel 41Cr4 was manifested in providing more favorable conditions for the functioning of the external DLC, increasing its adhesive bond strength to the substrate, reducing the stress level in the coating, and, as a result, the (CrAlSi)N/DLC was able to resist the existing heat and power loads a longer time.
The relevant passage is added.
Point 2: Details about lapping and polishing is required. How the parameters have been selected?
Response 2: Thank you for the remark. Additional processing of industrial ceramic inserts was carried out on a Lapmaster Wolters finishing and polishing machine with special finishing and polishing wheels using various diamond suspensions (grit of 50/40, 40/28 for finishing and 10/7, 5/3 for polishing) at a cutting speed of 3 m/s. Finishing has taken of 28 min, and polishing was of 16 min. Grinding and finishing modes were chosen based on literature data [35,37] and the authors' experience [9,10,16]. The relevant sentence is added.
Point 3: What could be the reasons for different wear behavior among group of curves (10 curves) in Figure 9(a) and (b).
Response 3: Thank you for a good question. It was the main reason to conduct this research, but we agree it needs to be better explained in the text. The surface layer of industrially produced inserts is replete with numerous defects that are stress concentrators, and when external loads are applied, the wear pattern is difficult to predict. Thus, the destruction of the cutting part can occur at any time (Figure 9a), while it can be like tearing out of entire conglomerates (as shown in Figure 1, option 3) When minimizing defects, i.e., repeated reduction of stress concentrators, the probability of destruction of the cutting part occurs mainly as a result of the gradual abrasion of the grains of the ceramic (Figure 1, option 1). The relevant passage is added.
Point 4: Why round cutting inserts have been selected?
Response 4: Thank you for pointing it out. The geometry of the cutting inserts was chosen from the tasks of providing curved surfaces for manufacturing gas turbine engine parts. The relevant sentence is added.
Point 5: It would be better if results are presented for more than one cutting speed and the variations should be discussed for the three cases.
Response 5: Thank you for the fair remark. It can be a valuable idea for further research. However, within this study, the approach is already unique; the one speed is chosen not to overload the article with data since the manuscript is already 20 pages. The relevant sentence is added.
Round 2
Reviewer 1 Report
The manuscript has been corrected.
Reviewer 2 Report
Thank you very much for revision of your paper. Revised article is suitable for publication in present form.